# Adrenocortical Carcinoma in Childhood: A Systematic Review

**DOI:** 10.3390/cancers13215266

**Published:** 2021-10-20

**Authors:** Maria Riedmeier, Boris Decarolis, Imme Haubitz, Sophie Müller, Konstantin Uttinger, Kevin Börner, Joachim Reibetanz, Armin Wiegering, Christoph Härtel, Paul-Gerhardt Schlegel, Martin Fassnacht, Verena Wiegering

**Affiliations:** 1Department of Pediatric Hematology, Oncology and Stem Cell Transplantation, University Children’s Hospital, University of Wuerzburg, Josef-Schneiderstr. 2, 97080 Wuerzburg, Germany; Riedmeier_M@ukw.de (M.R.); imme.haubitz@gmx.de (I.H.); Haertel_C@ukw.de (C.H.); Schlegel_p@ukw.de (P.-G.S.); 2Department of Pediatric Oncology and Hematology, Medical Faculty, University Children’s Hospital of Cologne, 50937 Cologne, Germany; boris.decarolis@uk-koeln.de; 3Department of General, Visceral, Transplantation, Vascular and Pediatric Surgery, University Hospital, University of Wuerzburg, Oberduerrbacherstr. 6, 97080 Wuerzburg, Germany; mueller_s27@ukw.de (S.M.); konstantin@uttinger.com (K.U.); boerner_k1@ukw.de (K.B.); Reibetanz_J@ukw.de (J.R.); Wiegering_A@ukw.de (A.W.); 4Department of Biochemistry and Molecular Biology, University of Wuerzburg, Am Hubland, 97074 Wuerzburg, Germany; 5Comprehensive Cancer Centre Mainfranken, University of Wuerzburg Medical Centre, Josef-Schneiderstr. 2, 97080 Wuerzburg, Germany; Fassnacht_M@ukw.de; 6Department of Medicine, Division of Endocrinology and Diabetes, University Hospital, University of Wuerzburg, Oberduerrbacherstr. 6, 97080 Wuerzburg, Germany

**Keywords:** pediatric adrenocortical cancer, pediatric adrenocortical adenoma, pediatric adrenocortical tumor, prognostic factors, therapy

## Abstract

**Simple Summary:**

Pediatric adrenocortical tumors are rare. Little information is available on the incidence, risk factors, prognostic factors, treatment, and overall survival. In this systematic review, we performed a search of the current literature. The most common reported risk factors are age > 4 years, high pathological tumor score, and advanced stage in which prognosis is poor. Treatment options are surgery, radiation, or chemotherapy, but ongoing randomized trials are lacking. International prospective studies must be the next step to implement standardized clinical stratifications and risk-adapted therapeutic strategies.

**Abstract:**

Adrenocortical tumors are rare in children. This systematic review summarizes the published evidence on pediatric adrenocortical carcinoma (ACC) to provide a basis for a better understanding of the disease, investigate new molecular biomarkers and therapeutic targets, and define which patients may benefit from a more aggressive therapeutic approach. We included 137 studies with 3680 ACC patients (~65% female) in our analysis. We found no randomized controlled trials, so this review mainly reflects retrospective data. Due to a specific mutation in the TP53 gene in ~80% of Brazilian patients, that cohort was analyzed separately from series from other countries. Hormone analysis was described in 2569 of the 2874 patients (89%). Most patients were diagnosed with localized disease, whereas 23% had metastasis at primary diagnosis. Only 72% of the patients achieved complete resection. In 334 children (23%), recurrent disease was reported: 81%—local recurrence, 19% (*n* = 65)—distant metastases at relapse. Patients < 4 years old had a different distribution of tumor stages and hormone activity and better overall survival (*p* < 0.001). Although therapeutic approaches are typically multimodal, no consensus is available on effective standard treatments for advanced ACC. Thus, knowledge regarding pediatric ACC is still scarce and international prospective studies are needed to implement standardized clinical stratifications and risk-adapted therapeutic strategies.

## 1. Introduction

In childhood, three different tumors arise from the adrenal gland: neuroblastic tumors, pheochromocytoma, and adrenocortical tumors (ACT). Neuroblastoma and pheochromocytoma originate from the adrenal marrow, whereas ACT develop from the adrenal cortex. With an estimated incidence of 1 per 1,000,000 children each year, ACT account for less than 0.2% of all pediatric malignancies [1]. Nevertheless, ACT present as a unique entity with distinct features. Both benign and malignant ACT are often hormonally active and present clinically with excessive hormone production and typical clinical syndromes [2]. Adrenocortical adenoma (ACA) accounts for approximately 20% of pediatric ACT and is associated with excellent prognosis. In contrast, adrenocortical carcinomas (ACC) exhibit aggressive behavior and, in advanced stages, require multimodal, multidisciplinary treatment. The clinical behavior and biological principles of pediatric ACT seem to be different from adult ACC [3,4,5,6,7] as young patients (<4 years) have better outcomes than adults. Some authors attribute these findings to an overestimation of malignancy based on the use of adult scores, which may not be adequate for the pediatric population. Other studies have discussed a different pathogenesis of prepubertal ACC [8,9]. Overall, the pathogenesis is not completely understood [10]. A subgroup of pediatric ACC is frequently associated with Li Fraumeni syndrome (LFS), a familial genetic cancer predisposition caused by a germline mutation in the TP53 tumor suppressor gene [11]. Thus far, no international consensus has been reached on the best therapeutic approach to ACC beyond surgery, which is the first-line treatment in most patients. Chemotherapy including mitotane has been reported for advanced stages, but the standardized implementation of druggable targets has not been described yet.

The main objective of this systematic review was to summarize the published evidence on pediatric ACC. We focus on the clinical and pathological characteristics, risk factors, and treatment modalities in order to provide an overview of the current literature and identify knowledge gaps and potential areas for future research.

## 2. Methods

In order to find suitable publications for our systematic review, we searched the PubMed and Embase databases up to 15 February 2021 for the manuscripts published after 1 January 1986. The exact search strategy for PubMed was as follows: (“child*”[Title/Abstract] OR “pediatric*”[Title/Abstract]) AND (“cancer”[Title/Abstract] OR “carcinoma”[Title/Abstract] OR “tumor”[Title/Abstract] OR “malign*”[Title/Abstract]) AND (“adrenocortical”[Title/Abstract] OR “acc”[Title/Abstract] OR “adrenal*”[Title/Abstract]); for Embase, it focused on the title and the abstract: (“child*” OR “pediatric*” AND (“cancer” OR “carcinoma” OR “tumor” OR “malign*”) AND (“adrenocortical” OR “acc” OR “adrenal*”).

All types of studies with abstracts available in German or English were included in the first step. Duplicates were automatically removed both by Endnote and manually. Two independent reviewers (M.R., V.W.) screened the titles and the abstracts of all the studies. Potentially relevant articles underwent full-text review to determine eligibility for inclusion in our analysis. Inclusion criteria were a minimum of three reported ACC patients younger than 21 years and reporting of clinical or pathological characteristics or treatment. Any disagreement on manuscripts was discussed and solved by consensus. Excerpted data were double-checked (M.R., V.W.). We performed this systematic review following the PRISMA guidelines and the selection process is visualized in the PRISMA flowchart in Figure 1 [12]. All the patients with ACT from the previously published studies were included in the analyses. If the same study population was examined in two different publications, we highlighted duplicate reporting of cohorts, but the reported cohorts were often not identical, and therefore double-counted patients from different studies are a limitation of our meta-analysis. The literature organization was performed using Endnote20. The charts and the tables were created with Microsoft Word and Microsoft PowerPoint. Statistical analyses were performed in the MEDAS software (Grund EDV, Margetshöchheim, Germany). Categorical variables were compared between the two groups using either the chi-squared or, when the values were expected to be small, Fisher’s or the Mehta and Patel exact test. Continuous measurements were compared between the two groups using the Mann–Whitney U test. Comparisons of more than two groups were performed using rank variance analysis according to Kruskal and Wallis; *p* < 0.05 was considered significant. In the Results section, portions and percentages were calculated according to the reporting studies.

## 3. Results

The database search identified 2961 articles. After removing duplicates, 2075 were left for further investigation. After screening the titles and the abstracts, 269 manuscripts appeared suitable for the purpose of our review. Full-text review of these 269 reports identified 137 manuscripts that matched the inclusion criteria; they were included in our analysis (Figure 1). None of these reports were randomized controlled trials (RCTs), mostly retrospective analyses. The articles were grouped into different categories according to their research topic (Table 1).

### 3.1. Incidence

According to the American National Cancer Institute, adrenocortical carcinomas account for approximately 0.2% of all childhood malignancies, and the majority are identified in the context of LFS [13,14,15]. An annual worldwide incidence of 0.3–0.38/million children < 15 years of age has been reported [16]. Of note, in southern Brazil, the incidence of pediatric ACC is 10–15 times higher than the worldwide occurrence [17,18]. This is thought to be related to a specific germline mutation at codon 337 (c.1010G > A, p.Arg337His) in the TP53 gene [19,20], which is highly frequent within the Brazilian population. Pinto et al. [20] and Garritano et al. [21] demonstrated that these mutations have a founder effect on ACC development in Brazil. As this germline mutation has been detected in >80% of Brazilian patients with childhood ACT [22] and may have caused distinct disease features, we report the clinical characteristics separately for Brazilian and other pediatric ACC patients.

### 3.2. Clinical Characteristics

The patient characteristics (Table 2, Appendix A) were described in 37 publications regarding the Brazilian population [2,17,22,23,24,25,26,27,28,29,30,31,32,33,34,35,36,37,38,39,40,41,42,43,44,45,46,47,48,49,50,51,52,53,54,55,56] and in 57 manuscripts regarding the patients in other countries [1,3,4,7,9,13,17,56,57,58,59,60,61,62,63,64,65,66,67,68,69,70,71,72,73,74,75,76,77,78,79,80,81,82,83,84,85,86,87,88,89,90,91,92,93,94,95,96,97,98,99,100,101,102,103,104,105,106].

#### 3.2.1. Brazilian Cohort

Within the reported Brazilian cohort (Table 2, Appendix A), there were two additional related cohorts from Argentina with the same mutation spectrum [28,52] and one cohort excluding the typical Brazilian mutation [47]. All of the other reports, if mentioned, documented an incidence of 80–100% for the specific germline mutation at codon 337 (c.1010G > A, p.Arg337His). The Brazilian cohort comprised 1656 patients (70% female) diagnosed between 1950 and 2019. The median age at diagnosis was 3.3 years (range: from 1 day to 17.3 years). The median time between the first symptoms and diagnosis was 6.8 months. Follow-up periods ranged between 1.8 and 10 years. Hormone activity was evident in a median of 97% of the patients (55%—androgens only, 30%—mixed, 3%—glucocorticoids only; deviations can be explained by the rounding and representation of the median). Most of the studies did not systematically report the overall survival. Nineteen percent of the patients died of disease, and the overall survival was reported to be between 34.6% and 95% and strongly depend on the stage distribution of the analyzed cohort. Relapse was reported in 144 patients (8.7%; corrected for the reporting studies, 144/625 = 23%), and secondary malignancies were described in two patients (0.1%; corrected for the reporting studies, 2/11 = 18%). However, these numbers may be underestimated because not all of the studies systematically reported relapses and secondary malignancies. The number of secondary malignancies is explained by the Li-Fraumeni/Li-Fraumeni-like TP53 mutation present in almost all the patients of this cohort. In addition, the patients may have been included in more than one report. Therefore, several features of the patient characteristics may be overestimated. Overlapping cohorts were mentioned in the following reports: Bergada et al. and Venara et al. [28,52]; Mendonca et al. and Latronico et al. [35,41]; Abduch et al., Leal et al., Leite et al., Lira et al., and Lorea et al. [22,23,33,37,38]; and in the non-Brazilian cohort (Appendix A).

#### 3.2.2. Non-Brazilian Cohort

The non-Brazilian cohort (Table 2 and Appendix A) comprised 2024 patients diagnosed between 1918 and 2020 (64% female). The median age was 5.1 years (range: from 1 day to 21 years). Time to diagnosis was a median of 6.0 months. Follow-up periods ranged between 2 and 12 years. Hormone activity was positive in 86% of the patients (50%—androgens only, 26%—mixed, 14%—glucocorticoids only). Most of the studies did not systematically report on the overall survival. Twenty-five percent of the patients died of disease, and the overall survival was between 14% and 100%, strongly depending on the stage distribution of the analyzed cohort. One-hundred-ninety relapses (9.4%; corrected for the reporting studies, 190/845 = 22%) and 28 secondary malignancies (1.4%; corrected for the reporting studies, 28/271 = 10%) were reported. Again, these numbers may be underestimated because not all the papers systematically reported on relapses and secondary malignancies.

The non-Brazilian cohort differed significantly from the Brazilian cohort, with a higher median age of diagnosis (*p* = 0.03), a lower proportion of female patients (*p* = 0.01), and a lower proportion of hormonally active tumors (*p* < 0.001). However, by excluding patients who were potentially double-reported and including only the most recent study, we found significant differences only for the higher proportion of hormonally active tumors (*p* < 0.01) in the Brazilian cohort. Relapse rate and stage distribution did not differ significantly between these groups.

#### 3.2.3. Age-Dependent Clinical Characteristics

A total of 40 studies reported detailed age-dependent patient characteristics (*n* = 1349) [2,3,4,7,13,22,25,26,28,30,31,35,40,41,45,48,49,57,62,63,64,65,69,70,72,74,76,77,80,81,83,84,86,89,90,91,94,97,98,103]. Comparing the Brazilian and non-Brazilian cohorts, even when excluding the potentially double-reported patients, we found a significantly different age distribution (*p* < 0.001). In the Brazilian cohort, 70% of the patients were <4 years old, 10%—>14 years old, whereas in the non-Brazilian cohort, 51% of the patients were <4 years old, 25%—>14 years old. In addition, more hormonally active tumors were observed in the Brazilian cohort (96% vs. 85%), with a predominance of androgen-producing tumors (55% vs. 42%) and fewer glucocorticoid-producing tumors (6% vs. 10%, respectively) than in the non-Brazilian cohort. We found no significant differences between the groups with regard to stage, overall survival, chemotherapy application, and sex distribution. Comparing the whole cohort regarding the age groups (<4, 4–14, and >14 years), we found significant differences (*p* < 0.001) regarding hormone activity and the kind of hormones produced, as well as death from disease, stage distribution, cancer predisposition syndromes (CPSs), and chemotherapy application (Table 3). There was no dependency in the sex distribution. In general, with increasing age, patients presented more frequently presented with hormonally inactive or only glucocorticoid-producing tumors, had a higher stage, received chemotherapy, and died due to disease.

### 3.3. Staging

The initial staging was reported for 2238 patients in 43 publications [2,3,9,13,22,26,29,30,32,33,36,37,38,39,42,44,45,46,47,48,50,51,53,54,56,59,62,70,71,76,77,80,81,82,93,94,95,96,98,99,101,106]. The median number of patients in each cohort was 44. Tumor stage was reported according to the international TNM classification system or the ENSAT criteria [107]. However, the criteria changed with time (Table 4) with regard to stage III and IV, so there may be minor bias in the advanced stages. Almost half of the reported patients (44%) were stage I, 25%—stage II, 13%—stage III, and 17%—stage IV (i.e., advanced stages were reported in about 30% of the cases). The details are shown in Table 2 as well as in Appendix A. No relevant differences were demonstrated between the Brazilian and non-Brazilian cohorts with regard to staging distribution. In general, the proportion of advanced stages increased with age at diagnosis as noted above. However, as most of the studies were retrospective, staging distribution was partly dependent on the focus of the study.

### 3.4. Metastasis

Information on metastasis was available in 69 reports [4,7,24,25,27,28,29,32,48,51,52,55,57,59,60,61,63,64,67,68,70,72,73,85,86,89,98,99,101,102,105,106]. Among the 2526 patients, at least one metastasis was reported in 632 (25%; 22%—at diagnosis, 3%—at relapse). Relapses were reported in 334 patients (of the 1470 patients, 23%). Of these relapsed patients, 65 (19%) presented with distant metastases, meaning that the majority of the relapses were local recurrences (Table 5). The distribution of metastases was described as follows: solely lymph nodes—52 patients, lungs—152 patients, liver—124 patients, both lungs and liver—22 patients, skeletal/bone marrow—17 patients, renal—14 patients, and CNS in 8 patients. Multiple abdominal locations were found in 26 patients, mostly peritoneal plus solid organ infiltration (liver or pancreas). The distribution of single-site metastasis was as follows: 52%—pulmonary, 43%—hepatic, 5%—skeletal, 4%—renal, and 2%—CNS. However, the invasion of lymph nodes is likely to be underestimated, as no specific details were available regarding surgery, lymph node exploration, and examination in most cases. Despite these limitations in reporting, this profile of metastatic distribution may best reflect the clinical reality of advanced stage manifestation, whereas relapses generally concerned either pulmonary metastases or multiple locations (see also Appendix A).

### 3.5. Pathological Characteristics

Regarding the pathogenesis of ACC, the pathways most involved in tumorigenesis are beta-catenin, insulin-like growth factors, p53/Rb signaling, and the chromatin remodeling process [108,109,110,111]. Pediatric (especially prepubertal) ACT probably derives from the fetal zone of the adrenal gland, which could explain the early age of onset and hormonal secretion, causing virilization due to androgen production [5,76,112]. Therefore, it is no surprise that pediatric adrenal cortical neoplasms appear to behave differently than histologically similar tumors in the adult population. Pathological criteria for malignancy in the adult population are well-established and routinely employed in clinical practice [113,114]. In contrast, definitive pathological criteria for malignancy in pediatric adrenal cortical neoplasms remain uncertain and are still subject to ongoing discussions.

The classic Weiss criteria [113], which are well-established in the adult population, are controversial in the pediatric literature. Even though multiple authors have shown that a Weiss score >3 correlates with poor prognosis [9,26,29,38,54,71,88,94,95,96,97,115,116,117] (Table 5 and Appendix A), others have shown that Weiss criteria are not applicable in the pediatric population [9,54,64,69,76,77,82,85,88,94,102,116] and modifications are necessary, as suggested by Wieneke et al. [9] (see Figure 2). There have been several studies validating the reliability of the Wieneke scoring system in both Brazilian and non-Brazilian cohorts of pediatric ACT [64,70,82], showing that it is highly predictive of the clinical outcome. In addition, vena cava invasion and periadrenal extension are 100% specific for the diagnosis of ACC [64], and focal myxoid stromal changes are highly distinctive for ACC [118].

For the detailed data on the factors of poor survival in pediatric ACC of the original publications identified by the systematic literature review (stage, pathological grading, tumor size, age >4 years, metastasis (M), lymph nodes (N), relapse, tumor extension (T), vascular invasion (V), venous (tumor) thrombosis (VTT), surgical outcome, hormonal activity, pathologic criteria, mitotic index), number of patients, number of articles, and short descriptions of the particular factors, see also Appendix A. Molecular markers that have been reported to be associated with a more aggressive disease course and limited outcomes are GATA4 overexpression at the protein and mRNA levels, especially in patients with metastases [25], SF1 and DAX1 immunohistochemical staining [119,120], IGF1R mRNA overexpression and immunohistochemistry [24,33,121], overexpression of Aurora kinase A (AURKA) and AURKB [26], MIR-149-3PIS [122] or HMGCR at the mRNA level [123], lower BCL2 and TNF expression at the mRNA and protein levels [38], chromosomal instability involving three or more chromosomes [124], and higher histone mRNA labeling indices [125]. Loncarevic et al. demonstrated a positive correlation between a high number of comparative genomic hybridization imbalances (>10) and fatal outcomes [81]. In addition, the following angiogenic markers are associated with poor prognosis: endoglin microvessel density (MVD) >1 mv/field, CD34 MVD <32 mv/field, and VEGF expression levels >4.8% on immunohistochemistry [32]. In addition, yes-associated-protein-1 (YAP1) overexpression at the protein and mRNA levels was demonstrated to be a marker of poor prognosis for pediatric ACT [23]. However, all of these associations are single-center observations that need to be confirmed in different settings. Additional histopathological markers that are more established and have been associated with prognosis, such as atypical mitosis, aneuploidies, tumor necrosis, Ki67, high mitotic index, and non-R0 resection, are described in the next section.

IGF2 overexpression on the protein and mRNA levels [111] has been described for pediatric ACC, although other analyses did not confirm this [103]. Furthermore, there are distinct molecular features of pediatric ACT that differ from adult ACT, specifically:(i)p53 variants on DNA examination [11,126] and p21 overexpression and abnormal beta catenin distribution at the mRNA level [103];(ii)higher proliferation index (immunohistochemistry) [125,127];(iii)increased expression of silver-binding nucleolar organizer regions (agNOR type III, immunohistochemistry) [127];(iv)placental alkaline phosphatase (PLAP) was detected by immunohistochemical analysis in one third of prepubertal ACC [128];(v)1p15 LOH as a widespread finding in pediatric ACT not related to malignancy [129].

De Sousa et al. showed that expression of DAX1, a stem cell fate regulator, was more frequent in pediatric ACC than in adult ACC at the protein and mRNA levels and revealed a significant positive correlation between DAX1 and SF1 expression, suggesting a potential role in tumorigenesis without relevance for prognosis [120]. In addition, human endostatin variant p.D104N has been described in DNA analysis to be more common in pediatric ACT (80.6%) [130]. Unlike in adults, the expression of matrix metalloproteinase 2 or the loss of HLA class II antigens based on immunohistochemical analysis does not discriminate between benign and malignant tumors in children [82]. Comparisons of the number and kind of genomic imbalances in children and adults have revealed characteristic differences. Gain of 1p and loss of 4p, 4q, and 16q are frequently found in children but are rare in adults. Inversely, loss of 1p is rare in children but is frequently found in adult ACT [81].

### 3.6. Molecular Changes as Potential Targets

As individualized targeted therapy is generally evolving in pediatric oncology, a few reports described potential targets derived from molecular analysis also for ACC. Notably, most of the findings are related to in vitro assays and have not yet been investigated along the translational research chain (Table 6). 

Directly or indirectly druggable targets of pediatric ACC of the original publications identified by the systematic literature review: target, intervention, in vitro data, case reports/studies. Borges et al. observed more aggressive disease in tumors with overexpression of AURKA and AURKB and presumed that the inhibition of these proteins could be a promising approach for the treatment for ACC [26] by using Aurora-specific degraders [138]. However, only in vitro data exist thus far [139,140]. Lira et al. [123] discussed IGF1R inhibition as a possible target, but the in vitro data were not convincing [33]. However, in two phase II/II studies, IGF pathway inhibition was well-tolerated and may have had some favorable effects on the outcomes [135,136]. Lin et al. [123] presumed a positive effect of lovastatin on tumors overexpressing HMGCR in vitro. Pianovski et al. suggested VEGFR1 and mTOR inhibitors for the treatment of pediatric ACC, demonstrating that almost all the patients overexpressed VEGFR and many patients had activation of the mTOR pathway [137]. However, there have been no convincing clinical reports on this approach [85,131,141,142]. In a network analysis comprising 18 pediatric ACC patients, CDK1, CCNB1, CDC20, and BUB1B were identified as potential biomarkers of pediatric ACC, though not as expressive as in adult cohorts [133], and as potential targets for a therapeutic approach [134]. Finally, Abduch et al. showed that in adrenocortical cells, there is a close crosstalk between YAP1 and Wnt/beta-catenin. These data allow the possibility of future molecular therapies targeting the Hippo/YAP1 signaling [23]. In the pan-genomic approaches [109,110,111], the ATRX and ZNRF3 genes were associated with adrenal tumorigenesis for the first time. The ATRX gene is responsible for chromatin remodeling and telomeric structure maintenance, and the initial reports indicated that somatic ATRX mutations are associated with poor event-free survival [111], whereas the ZNRF3 genes could not be associated with prognosis [143].

However, none of these markers can be considered established, and all of them are experimental. Systematic molecular analysis of pediatric ACC, especially outside of Brazil, is rare.

Regarding the immune response in pediatric ACT, two studies were found: Parise et al. reported an association between infiltrated CD8+ T cells and better prognosis [45], whereas Geoerger et al. demonstrated responses to checkpoint inhibition (pembrolizumab) in children [85,132]. Further functional analyses are necessary to clarify immune escape mechanisms of pediatric ACC to provide future targeted therapies.

### 3.7. Prognostic Factors

Prognostic outcome parameters were identified in 65 publications (Appendix A. The most common factors for poor overall survival were advanced tumor stage (TNM, modified ENSAT) [2,8,9,13,26,37,38,43,45,47,50,51,53,54,70,76,87,94,95,97,105,106,116,144,145,146], large tumor mass (g) [1,2,4,7,8,9,13,26,28,29,37,38,47,49,52,65,69,70,76,77,88,94,97,116,147], large tumor size [2,13,40,41,42,83,84], tumor volume (cm^3^) [1,4,7,9,13,26,29,37,38,49,50,51,52,53,57,65,69,70,75,76,77,78,94,95,97,102,145,147], and high pathological tumor score (Wieneke, Weiss, ENSAT) [9,26,29,38,45,54,71,88,94,95,96,97,115,116,117,145]. Furthermore, tumor extension [1,3,22,33,36,38,39,45,46,47,49,50,54,56,71,74,75,76,77,79,81,82,90,91,92,93,94,95,96,97], existence of metastasis (lymph nodes [29,57,75,87,94] or distant metastasis [29,57,68,75,76,79,87,94]), tumor relapse [1,29,44,57,65,69,75,76,77,94,105], vascular invasion [1,9,44,51,57,65,69,75,76,77,94], and venous thrombosis [44] have been described as negative prognostic factors.

Age <4 years at diagnosis has been described as a favorable factor by numerous authors [2,3,7,13,26,29,37,38,44,45,47,49,51,53,54,57,64,69,70,75,76,78,87,144,145,148]. Hormone activity was associated with both better [68] (virilization alone [2,28,53,84,145], non-Cushing [47,70]) and worse outcomes [4,28,54,57] (mixed [53,54]). In a comprehensive analysis of all the available data, we confirmed age and hormone activity as important prognostic factors (Table 5 and Appendix A)

Regarding the influence of the surgical approach on the outcomes, incomplete resection [2,29,43,45,57,61,68,70,71,72,74,75,77,83,84,87,94,97,99,146], biopsy [1,94], and tumor spillage [1,2,29,44,91,97] were associated with poor overall survival. Histopathological criteria associated with poor survival are atypical mitosis [9,28,65,77], aneuploidy [9,65,147] (no effect [7,29]), tumor necrosis [7,8,9,29,65,77,78,94], high Ki67 expression [4,47,54,65,69,71,94,125] (no effect [45]), and high mitotic index [7,9,28,65,69,77,78,94,147] (no effect [29,59]).

A recent meta-analysis [149] of 42 studies found the following predictors of a better outcome: age <4 years, non-secreting tumors, complete surgical resection, tumor volume <200 cm^3^, tumor weight <100 g, maximum tumor diameter <5 cm, and lower stage at diagnosis. However, these factors correlate with each other. For example, a lower stage means a smaller tumor volume, which is associated with better outcomes. The patients affected by Cushing syndrome had worse outcomes.

### 3.8. Therapy

Therapy regimens were described in 65 publications [1,2,3,4,7,9,13,22,28,29,32,36,39,40,41,43,45,47,48,49,50,51,52,56,57,59,60,61,62,63,68,70,71,72,74,75,76,77,78,79,80,82,83,84,85,86,88,90,91,92,93,94,95,96,97,98,99,103,106] (Table 2 and Appendix A). Most of them are retrospective cohort analyses; only one registry analysis was published during our screening interval (and another one—that is discussed in the Discussion section—outside of our screening interval) [17,95,150]. Therefore, therapies are hardly comparable, and adequate analysis of response rates and the influence of certain therapeutic elements (mitotane levels, chemotherapeutic agents, etc.) are limited. We identified a total of 2221 reported therapies, mostly in a retrospective setting. Most patients underwent surgery (*n* = 2036; 92%), followed by cytotoxic chemotherapy (*n* = 528; 24%), mitotane treatment (*n* = 360; 16%), and/or radiation (*n* = 69; 3.4%). Initial biopsy was performed in 69 patients (3.4%). Of the 1358 patients, 976 (72%) received an R0 resection, and in 69 patients (3.4%), intraoperative tumor spillage occurred. Of the 528 patients, 18 received (palliative) chemotherapy without a surgical approach. These data fit the reported stage distribution (stage III—13%, stage IV—17%) and the common practice of (neo-)adjuvant treatment in advanced stages.

### 3.9. Secondary Malignancies and Cancer Predisposition Syndromes

In southern Brazil, the high incidence of pediatric ACC is explained by a specific germline mutation at codon 337 (c.1010G > A, p.Arg337His) in the TP53 gene [19,20], but other cohorts are not systematically screened for CPSs. Eight publications [59,61,80,84,85,93,101,103] more systematically reported CPSs in the non-Brazilian cohorts, with frequencies ranging from 14% to 81% (Appendix A). Additional cases of CPS were reported in 11 other publications [7,57,71,72,74,78,79,81,97,98,99]. LFS is the most common cause of CPS related to ACC. Six patients with Beckwith–Wiedemann syndrome [72,85,97,98] were described in the literature. A total of 29 cases of secondary malignancies/neoplasia were described [7,40,48,60,61,66,74,79,84,92], including the following entities: astrocytoma (*n* = 4), osteosarcoma (*n* = 3), liposarcoma (*n* = 2), hepatoblastoma, chondrosarcoma, amelanotic melanoma, non-Hodgkin lymphoma, optic nerve glioma, kidney cell carcinoma, contralateral ACC, and choroid plexus papilloma (*n* = 1 each), as well as lipoma (*n* = 3) and hemangioma (*n* = 2) as benign neoplasias. Even if there are only sporadic reports, nationwide analyses and genomic profiles [11,151,152,153,154,155,156,157,158] hypothesize that the number of CPSs may be much higher than thought, and genetic counseling should be recommended to all parents and legal guardians of pediatric patients with ACC.

## 4. Discussion

Pediatric ACT is a rare disease, and many questions remain to be answered; 80–90% of these tumors are carcinomas (ACC) [2] which in young children present with less aggressive behavior and more frequently as hormone-positive tumors than their adult counterparts. Clinical and histopathological differences between pediatric and adult ACT suggest two distinctive and possibly different mechanisms of carcinogenesis. The first is that pediatric (prepubertal) ACT are thought to derive from the fetal zone of the adrenal gland, which could explain the early age of onset and hormonal secretion [5,76,112]. However, the pathogenesis remains unclear at this point and, due to the observed differences, the results of adult cohorts should not simply be transferred to pediatric ACT patients. In this systematic review, we summarized the published evidence for pediatric ACCs with a focus on clinical and pathological characteristics, therapy modalities, and risk factors.

From our personal perspective, the most critical unanswered questions resulting from this overview are:What are the best prognostic markers at the time of diagnosis?What is the impact of surgery and how can it be optimized?Are there alternative tumor markers (to 24 h urine samples) that can be used more easily for follow-up investigations?How can we improve the outcomes in pediatric patients at advanced stages?How can we reduce the relapse rate?Are there specific molecular targets that can be used for tailored therapies?

Looking at the reported data, a key limitation is the small sample sizes of the reported cohorts. Even with the efforts over the last few decades to improve therapies via network collaboration, such as the GPOH-MET registry (Germany), the International Pediatric Adrenocortical Tumor Registry (IPACTR, mainly parts of Brazil and some US centers) [150], and the EXPERT data (European Consortium of Rare Diseases) [16,57,95], most studies have been retrospective, and systematic clinical and molecular analyses of clinical improvement among patients in therapeutic studies are still lacking. Recently, Rodriguez-Galindo et al. published results of the ARARO332 protocol [150], which underline the significant influence of stage and age on the outcome. They further demonstrated that the relapse rate is high in locally advanced tumors and may be improved by mitotane and/or chemotherapy.

Comparing pediatric relapse rates to those of adults, it appears that the rate of local recurrence is relatively high. Looking at our systematic approach, a total of 334 relapses were reported in 1470 patients (23%). This number may be underestimated because follow-up periods and reporting quality differed between the reports. However, if we compare the number of reported metastases at relapse to the overall number of reported relapses, we found 65 cases (19%) with distant metastases in a total of 334 reported relapses. This means that 81% (269/334) of relapses are due to local recurrence or lymph node metastases. Local recurrence in the adult setting has been reported to be 50–60% [159]. This fact may allow improvement of patient outcomes due to advanced local therapy, extended surgery, and/or radiotherapy. Recent data (on adult cohorts) show that primary lymphadenectomy [160,161], as well as the oncological experience in ACT [162], have an impact on the outcome. In our analysis, pediatric surgeons most often (if reported) performed surgery on pediatric patients, and the centers’ experience with ACT surgery was limited when looking at the reported time intervals and patient numbers [56,150]. One step towards therapy optimization may be the systematic retrospective (and prospective) analysis of the surgical procedures performed regarding lymphadenectomy, relapse rates, and surgical experience to define subgroups with an increased risk of relapse and that will benefit from these approaches. The ARARO332 trial already tried to address this question and could not find a benefit for survival. However, the number of removed lymph nodes was mostly low, and primary lymphadenectomy is not a commonly performed procedure in pediatric patients, which may explain the low compliance and contrasting results. This phenomenon has also been reported in paratesticular rhabdomyosarcoma [163]. Especially for this specific patient cohort, it is mandatory to advocate for centralization of surgical procedures, which is beneficial in the vast majority of oncological surgical procedures [164,165,166,167], particularly for rare entities.

In addition to the surgical therapy, the possible impact of radiotherapy on local recurrence has to be evaluated further in pediatric patients. In the adult setting, adjuvant tumor bed irradiation has been shown to be effective in reducing the high rate of local recurrence in ACC [168]. Two recent meta-analyses regarding radiotherapy in ACC patients have independently shown reduced local recurrence-free survival [169,170]. Effects of better overall and relapse-free survival have also been shown [170]. In the pediatric setting, improvement of overall and disease-free survival due to local radiotherapy is known for several tumors, including neuroblastoma [171]. Over the last few decades, the ENSAT consortium [172,173], which mainly works with adult ACT patients, has shown how centralization and networking can continuously improve patient outcomes, even in rare cancer entities, and should serve as a model for pediatric ACC.

The third issue is the histopathology of pediatric ACC. As we described, there are different pathological features that have been associated with the outcome. In addition, there are differences in the expression profiles and predictive markers in adult ACT. At this point, the etiology and tumorigeneses of pediatric ACT remain unclear and have to be evaluated further.

By analyzing age-dependent influence on clinical characteristics and outcomes, we could confirm that younger patients have a different profile of hormone production and present with limited disease more often than older pediatric patients. Comparing the non-Brazilian and Brazilian cohorts, we found that the children were younger in the Brazilian cohort. However, stage distribution, outcome, and hormone activity did not differ between the groups. Therefore, the higher number of hormonally active tumors, which is widely described in the literature for the Brazilian cohort [15,145,174], may not be due to another tumor subgroup due to the specific germline mutation, but may be an age-dependent effect [11]. The presence of a germline mutation has not been systematically investigated. However, if one examines the reported cases, a clear predominance in infancy is noticeable, even in the non-Brazilian cohort. There are different hypotheses regarding the pathogenesis of childhood ACC. One hypothesis states that early childhood tumors arise from the fetal zone of the adrenal gland, whereas another states that the onset of puberty and associated hormone change has an important influence [108,111,175]. There are several limitations to our evaluation due to the retrospective character and broad time interval of the reporting studies. Nevertheless, the differences between early childhood (<4 years) and older pediatric patients seems more impressive than the differences between pre- and post-pubertal patients. This phenomenon is well-known from other pediatric tumors of the adrenal gland and underlies the hypothesis of fetal zone-derived tumorigenesis in early childhood [176]. The different stage distribution and hormone activity and the associated better prognosis could be due to differing tumor development. In addition, they seem to have a very high association with CPSs. To confirm the clinical risk factors, a prospective approach will be necessary to adjust for common risk stratification. In addition, further investigations will need to evaluate whether the tumor pathogenesis in older children is comparable to adult ACC.

Regarding therapy, there have been no randomized studies, and the reported therapeutic approaches are hardly comparable. A further limitation is the large time span in which the patients were treated. Most authors reported chemotherapy and surgery in advanced stages (stage III and IV) and a solely surgical approach only in early stages (stage I and II). These data fit the distribution of advanced stages, even if reports slightly differ. The indication for irradiation was the curative and palliative setting; therefore, the role of irradiation in pediatric ACC remains unclear. The rate of mitotane treatment in advanced stage pediatric ACC is low (16%), which likely correlates with the wide time interval and limited drug experience, especially at a younger age. For the first time, standardized therapy recommendations were published in 2012 on the basis of the German GPOH-MET registry data [1,95]. Recently, there have been efforts to publish international standardized diagnostic and therapeutic recommendations [150,177].

Vincristine, ifosfamide, Adriamycin, carboplatin, etoposide, and mitotane are well-established drugs in adult ACC and are also administered in the pediatric setting. The randomized FIRM-ACT trial [178,179] and ADIUVO (not yet published) show that chemotherapy and optimized regimens improve patient outcomes. Despite these results, there are still patient subgroups with limited prognosis, especially in pediatric stage IV patients. In the last few years, classic chemotherapeutic drugs could not achieve further improvement for these patients. Better molecular understanding may allow better discrimination and stratification of the patient subgroups in addition to the classic scoring systems (TNM, Wieneke, Weiss) [9,77]. Furthermore, it may identify new druggable targets, which will be a step towards more tailored therapy. Consequently, we should initiate randomized therapy protocols and central biobanking to advance our knowledge regarding the molecular basis of pediatric ACT and therapy in the international context.

Lastly, the association of pediatric ACT with CPSs is high, and may be even higher with advanced understanding of CPSs and awareness from the treating physicians. Standardized screening for CPSs is recommended for patients with pediatric ACC, as the frequency of LFS is up to 50% [180], but numbers were much lower in our systematic analysis outside the Brazilian cohort, and CPSs may be underestimated. In this context, treatment modalities, such as radiation therapy, and follow-up care need to be discussed critically. In Southern Brazil, the impact of neonatal screening for the common germline TP53 mutation R337H has been shown to be effective in detecting ACT when readily curable [146]. Therefore, we should offer all of our patients genetic screening after counseling the patients and parents. Even if the reported secondary malignancy rate for CPS appears to be low, the number of unreported cases is certainly high, as most ACC patients have an early age of onset, short follow-up, and no systematic recording in the analyzed studies. This becomes clear when considering that only two cases of secondary malignancy were reported in the Brazilian cohort. In view of today’s knowledge of CPSs and pediatric ACC [11], genetic counseling and individualized preventive care seem to be inevitable and may have significant implications for treatment in the future.

This study has some limitations. Not all articles that were suitable based on abstract screening were available as full text. We included them in our analysis if adequate data were available from the abstract. Furthermore, most suitable studies were performed retrospectively, which could have led to publication bias. The majority of the studies included a low number of patients, as mentioned in the tables.

## 5. Conclusions

In conclusion, our knowledge regarding pediatric ACC is still incomplete and needs to be advanced by an international effort. This review offers an overview of the published evidence for pACC. However, many questions have not been answered, especially when compared to adult ACT and other pediatric tumor entities, such as ALL or neuroblastoma. As pediatric ACT is a rare disease, only an international network with all pediatric (and adult) ACT specialists sharing all experience and data will be able to advance knowledge and therapies in order to improve the outcomes of ACT patients. The next and urgent step will be the development of biological tumor markers, molecular understanding, and tailored therapeutic concepts.

As shown for other tumor entities, other than the improvement of local therapy (surgical techniques and radiotherapy), the big challenges will be to achieve better understanding of molecular pathomechanisms and pathway activation. Scientific research programs on a collaborative basis, including methylation assays and genetic murine models, will be one step towards more tailored therapy options, especially in advanced stages. In the long term, the establishment of easy, feasible, and reproducible biological tumor markers, as well as liquid biopsies, would be another goal for improved monitoring, risk stratification, and follow-up. To the best of our knowledge, there are no established models for pediatric ACT. Our recommended approach and treatment algorithm for an integrated strategy is shown in Figure 3. International prospective studies are the next step to establish standardized clinical stratifications to improve adapted therapeutic strategies and streamline basic research questions for advanced understanding of the pathogenesis of pediatric ACC.

## Figures and Tables

**Figure 1 cancers-13-05266-f001:**
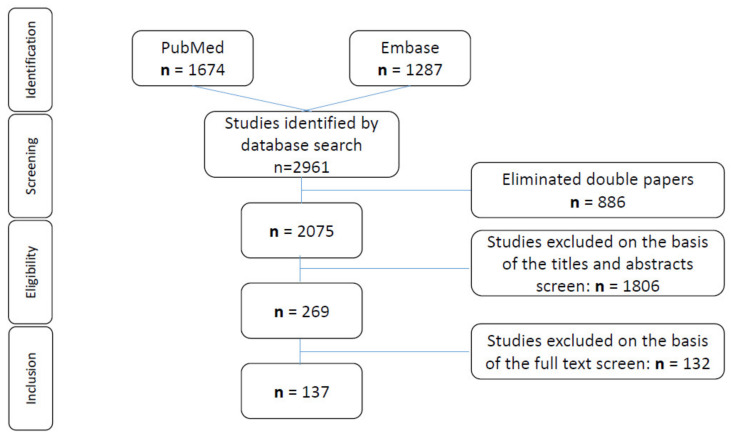
Search criteria. Flow diagram of the search strategy and evidence acquisition in a systematic review on adrenocortical carcinoma in childhood.

**Figure 2 cancers-13-05266-f002:**
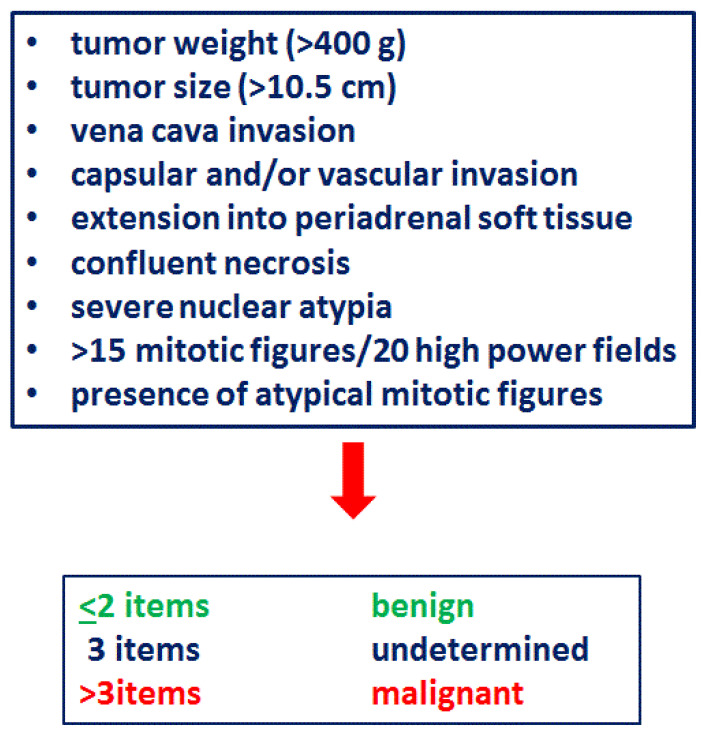
Wieneke criteria.

**Figure 3 cancers-13-05266-f003:**
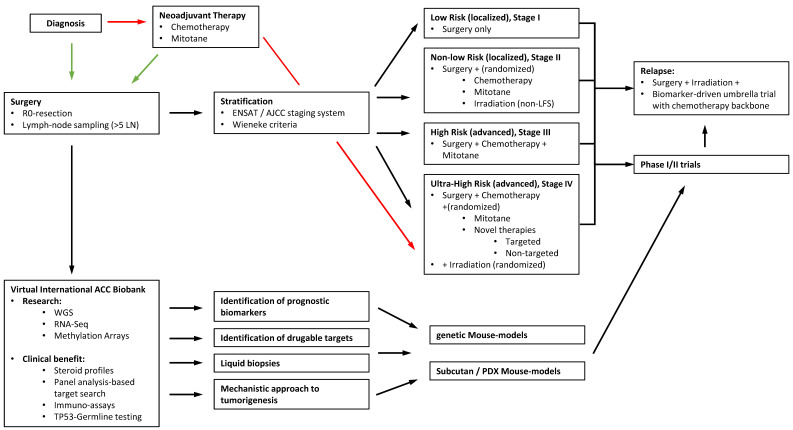
Recommended approach and treatment algorithm for a possible study design. Standardized consensus guidelines on surgery, stratification, and further treatments lead to randomized trials and are embedded in an integrative strategy including modern biobanking allowing preclinical in vitro and murine model studies as well as a set-up of phase I/II trials for novel targeted therapies.

**Table 1 cancers-13-05266-t001:** Aspects of research. Analyzed topics and numbers of the identified articles.

Aspect	Number of Identified Articles
Clinical characteristics, relapses, follow-up	94
Age-dependent clinical characteristics	40
Tumor stage at diagnosis	43
Metastasis	69
(Histo-)pathological characteristics	31
Druggable targets	11
Prognostic factors	65
Treatment modalities	65

**Table 2 cancers-13-05266-t002:** Summary of the data of the pediatric patients with an adrenocortical tumor from the selected original publications identified by systematic literature review regarding clinical characteristics, treatment details, and tumor stage distribution.

**Cohort**	**Clinical Characteristics**											**Number of Included Studies**
	** *n* **	**Years of age**	**%**	**%**	**%**	**%**	**%**	**%**	**Time**	**Time (m)**	** *n* ** **Relapses**	
	**ACT**		**Female Patients**	**Hormonally Active Tumors**	**Mixed**	**Androgens**	**Glucocorticoids**	**DOD**	**Interval**	**Symptoms–Diagnosis**		
Brazilian cohort	1761	3.3	70	97	30	55	3	19	1950–2019	6.8	144	37
Non-Brazilian cohort	1919	5.05	64	86	26	50	14	25	1950–2020	6.00	190	57
	**Treatment Details**											
	**Number of** **Patients**	**Surgery**	**R0**	**>R0**	**No Information on the Extent of Surgery**	**Tumor Spillage**	**Biopsy**	**Chemotherapy + Surgery**	**Only Chemo-Therapy**	**Radiotherapy**	**Mitotane**	59
Combined cohort (*n*)	2221	2036	976	235	1009	69	69	510	18	74	360	
(%)	100	92	72	17	50	3	3	24	3	3	16	
	**Tumor Stage Distribution**											
	**Number of** **Patients**	**I**		**II**	**III**		**IV**					48
Combined cohort (*n*)	2238	985		568	287		371					
(%)	100	44		25	13		17					

Clinical characteristics of the Brazilian (*n* = 1761) and non-Brazilian (*n* = 1919) cohorts: numbers of ACT and ACA patients, age, % of female patients, % of hormonally active tumors (% mixed, % androgens, % glucocorticoids), %DOD, time interval, time from symptoms to diagnosis (months), relapses, reported cases of CPS, country of population (see also Appendix A for the detailed data). Treatment details of 2221 patients: radiotherapy, mitotane, chemotherapy and surgery, surgery (R0, >R0), only chemotherapy, tumor spillage, biopsy (see also Appendix A for the detailed data). Tumor stage distribution of 2238 patients according to the international TNM classification system (stage I, II, III, IV, unknown/ACA; see also Table 4 for the stage definition and Appendix A for the detailed data).

**Table 3 cancers-13-05266-t003:** Age-dependent characteristics of the 1349 pediatric patients with an adrenocortical tumor from the selected original publications identified by the systematic literature review.

**Characteristics**		**Age**						
		**0–4 years**		**4–14 years**		**>14 years**		
		** *N* **	**%**	** *n* **	**%**	** *n* **	**%**	** *p* **
Gender	Female	285	65.22%	98	60.12%	91	70.00%	
	Male	152	34.78%	65	39.88%	39	30.00%	0.21
Hormonal activity	No	23	6.93%	12	9.45%	18	23.08%	
	Mixed	113	34.04%	34	26.77%	31	39.74%	
	androgens	180	54.22%	68	53.54%	13	16.67%	
	glucocorticoids	16	4.82%	13	10.24%	16	20.51%	<0.00000005 ***
DOD	Yes	53	13.98%	73	45.06%	82	52.23%	
	No	326	86.02%	89	54.94%	75	47.77%	<0.00000005 ***
Stage	I	209	58.87%	53	37.06%	31	25.00%	
	II	78	21.97%	32	22.38%	23	18.55%	
	III	44	12.39%	21	14.69%	14	11.29%	
	IV	24	6.76%	37	25.87%	56	45.16%	<0.00000005 ***
P53/CPS	Yes	98	82.35%	26	72.22%	5	35.71%	
	No	21	17.65%	10	27.78%	9	64.29%	0.0013 **
Chemotherapy	Yes	53	24.54%	38	55.07%	30	55.56%	
	No	163	75.46%	31	44.93%	24	44.44%	0.0000001 ***
***n* = 1312, 37 patients excluded because of inexact information on age (>4 years)**								
**Characteristics**		**Age**						
		**0–4 years**			**>4 years**			
		** *N* **	**%**		** *n* **	**%**		** *p* **
Gender	Female	285	65.22%		193	64.77%		
	Male	152	34.78%		105	35.23%		0.90
Hormonal activity	No	23	6.93%		31	14.76%		
	Mixed	113	34.04%		68	32.38%		
	Androgens	180	54.22%		82	39.05%		
	glucocorticoids	16	4.82%		29	13.81%		0.0000088 ***
DOD	Yes	53	13.98%		160	49.38%		
	No	326	86.02%		164	50.62%		<0.00000005 ***
Stage	I	209	58.87%		84	30.88%		
	II	78	21.97%		56	20.59%		
	III	44	12.39%		36	13.24%		
	IV	24	6.76%		96	35.29%		<0.00000005 ***
P53/CPS	Yes	98	82.35%		31	62.00%		
	No	21	17.65%		19	38.00%		0.0057 **
Chemotherapy	Yes	53	24.54%		68	55.28%		
	No	163	75.46%		55	44.72%		<0.00000005 ***
*n* = 1349								

Age cohorts (top (*n* = 1312): 0–4 years, 4–14 years, >14 years; bottom (*n* = 1349): 0–4 years, >4 years), age-dependent patient characteristics: gender (female, male), hormonal activity (no, mixed, androgens, glucocorticoids), tumor stage (I–IV), P53/cancer predisposition syndrome (CPS) (yes, no), chemotherapy (yes, no); total and percentage; ** *p* < 0.005, *** *p* < 0.0005.

**Table 4 cancers-13-05266-t004:** Changes in TNM classification over time.

Stage	UICC/WHO 2003	ENSAT 2008	UICC 2020 (since 2010)	AJCC, 8th Edition
I	T1, N0, M0	T1, N0, M0	T1, N0, M0	T1, N0, M0
II	T2, N0, M0	T2, N0, M0	T2, N0, M0	T2, N0, M0
III	T1–2, N1, M0 T3, N0, M0	T1–2, N1, M0 T3–4, N0–1, M0	T1–2, N1, M0 T3–4, N0–1, M0	T3, N0, M0 T1/2, N1, M0 T4, N0, M0 T3/4, N1, M0
IV	T1–4, N0–1, M1 T3, N1, M0 T4, N0–1, M0	T1–4, N0–1, M1	T1–4, N0–1, M1	T1–4, N0–1, M1

Definition of tumor stages: T1—tumor ≤ 5 cm; T2—tumor > 5 cm; T3—tumor infiltration in the surrounding tissue; T4—tumor invasion in the adjacent organs (ENSAT: also venous tumor thrombus in the vena cava/renal vein); N0—no positive lymph nodes; N1—positive lymph node(s); M0—no distant metastasis; M1—presence of distant metastasis.

**Table 5 cancers-13-05266-t005:** Factors of poor survival.

Factors of Poor Survival		Number of Patients	Number of Articles	Description
Advanced stage		1149	23	A higher tumor stage is associated with poor survival
Pathological grading: Weiss > 3, Wieneke > 3, ENSAT3/4		658	16	High pathological tumor score is associated with poor survival
Tumor size >100 g		1083	24	Large tumor mass is associated with poor survival
Tumor volume >200 cm^3^		1289	27	Large tumor volume is associated with poor survival
>4 years old		1260	24	Age >4 years is associated with worse outcomes
Metastasis	Distant metastases	627	8	Existence of metastases (lymph nodes and distant metastases) and tumor relapse are described as negative prognosis parameters
	Lymph nodes	416	5	
Relapses		27	1	
Tumor extension, vascular invasion, and/or venous thrombosis		779	13	Tumor extension, vascular invasion, and venous thrombosis are associated with worse outcomes
Surgical outcome:	Non-R0	636	12	Non R0-resection, biopsy, and tumor spillage are described as negative prognostic markers
	Biopsy	154	2	
	Tumor spillage	245	6	
Hormone activity		110	2	Hormone activity was associated with both better and worse outcomes
Immunohistochemistry	Atypical mitosis	137	4	Atypical mitosis, aneuploidy, tumor necrosis, high mitotic index with high Ki67 expression are associated with poor outcomes
	Aneuploidy	227	6	
	Tumor necrosis	360	8	
	High mitotic index (MI),	410	11	
	Ki67	348	9	

**Table 6 cancers-13-05266-t006:** Druggable targets.

Target	Intervention	In Vitro Data	Case Reports/Studies	References
Crosstalk between YAP1 and Wnt/beta-catenin	Hippo/YAP1 signaling inhibition	yes	none	Abduch et al., 2016 [23]
PDL1 expression	Checkpoint inhibition	yes	case reports/several trials	Altieri, 2020, Geoerger, 2020 [131,132]
Overexpression of Aurora kinases A	Aurora kinase inhibition	yes	none	Borges, 2013 [26]
CDK1, CCNB1, CDC20, and BUB1B		yes	none	Fragoso, 2012; Kulshrestha, 2016 [133,134]
HMGCR-overexpressing tumors	Lovastatin	yes	none	Lin, 2010 [123]
Overexpression of IGF1R	IGF1R inhibition	yes	phase ½	Lira, 2016; Jones, 2015; Weigel, 2014 [33,135,136]
Overexpression of VEGFR	VEGFR inhibition/tyrosine kinase inhibition	yes	case reports/several trials	Pianovski, 2013, Altieri, 2020 [131,137]
mTOR kinase activity	mTOR inhibition	yes	case reports/several trials	Pianovski, 2013, Altieri, 2020
ATRX		yes	none	Pinto, 2015 [111,131,137]
ZNRF3		yes	none	Pinto, 2015 [111]

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
