# Peer review of "Adrenocortical Carcinoma in Childhood: A Systematic Review"

_cancers, 2021, doi:10.3390/cancers13215266_

Round 1
Reviewer 1 Report
Riedmeier et al provide a comprehensive and interesting overview of the current literature on pediatric ACC.
Major comment
Although the paper covers a large amount of published data, it lacks structure. This has to be improved. According to the summary and introduction the aim of the paper is to summarize the published evidence. Defining which patients may benefit from a more aggressive therapeutic approach is also a goal. This latter goal is not addressed in the text. The research questions that are used to reach the goals are not mentioned in the introduction. The full search strategies for the systematic review(s) are missing.
The conclusion of the paper turns out to be a proposal for a multinational ACC initiative. Many of the components of this proposal -methylation, liquid biopsies, PDX models- are not even mentioned once in the text of the manuscript. The goals of the review, methods, results and recommendations for follow-up studies have to be better aligned.
It is my impression that the aim of this paper is to give an overview of the current literature, but more importantly identifity knowledge gaps and potential areas for future research using a systematic review approach.
Minor comments
Where in the manuscript are relative risk curves used (line 103)?
None of the tables -except table 2c- contain actual results and could be added as supplemental information
In section 3.1.5 is not clear whether molecular markers are DNA, RNA or protein, chomosomal loci are listed without explanation, please elaborate.
Author Response
Response to reviewers’ comments
Dear reviewers,
thank you all for your considerations and recommendations that greatly helped us to improve the quality of our manuscript for publication in Cancers.
Please find detailed response to your comments highlighted in the text below.
Review 1
Riedmeier et al provide a comprehensive and interesting overview of the current literature on pediatric ACC.
Major comment
Although the paper covers a large amount of published data, it lacks structure. This has to be improved. According to the summary and introduction the aim of the paper is to summarize the published evidence. Defining which patients may benefit from a more aggressive therapeutic approach is also a goal. This latter goal is not addressed in the text. The research questions that are used to reach the goals are not mentioned in the introduction. The full search strategies for the systematic review(s) are missing.
It is my impression that the aim of this paper is to give an overview of the current literature, but more importantly identify knowledge gaps and potential areas for future research using a systematic review approach.
Thank you very much for your valuable comments.
The goal that was defined in the introductory part is a long-term goal, while the goal of the systematic review was to identify knowledge gaps and potential areas for future research as a first step. We apologize this misleading introduction section. We changed this paragraph as highlighted in the manuscript.
The search strategies are described in the method section as well as in figure 1. However, as suggested, we added a link in the introduction section as highlighted.
The conclusion of the paper turns out to be a proposal for a multinational ACC initiative. Many of the components of this proposal -methylation, liquid biopsies, PDX models- are not even mentioned once in the text of the manuscript. The goals of the review, methods, results and recommendations for follow-up studies have to be better aligned.
Yes, you are right – we tried to improve this point as highlighted in the text and hope this have improved the quality of the manuscript
Minor comments
Where in the manuscript are relative risk curves used (line 103)?
We apologize this mistake, we planned, but not included relative risk curves in the manuscript. Therefore we delete this sentence.
None of the tables -except table 2c- contain actual results and could be added as supplemental information
We see your point and have added most of the information of the tables to supplemental material; the key messages are represented in new tables as you can see within the manuscript. Following changes are made:
- Tables 1/2c/3/6 and Figures 1-3 received minor changes and are now entitled Table 1 (remained Table 1), Table 3 (previously Table 2c), Table 4 (previously Table 3), and Table 6 (remained Table 6).
- Tables 2a and b/ 4/ 5/ 7/ 8 have been moved to the supplementary material and are now entitled Suppl. Table 1 a and b (previously Table 2 a and b), Suppl. Table 2 (previously Table 4), Suppl. Table 3 (previously Table 5), Suppl. Table 4 (previously Table 7), and Suppl. Table 5 (previously Table 8).
- Contents of (old) Table 2a and b, Table 4, and Table 8 are now summarized in the new Table 2 and summarized data of Table 7 can be found in the new Table 5.
In section 3.1.5 is not clear whether molecular markers are DNA, RNA or protein, chomosomal loci are listed without explanation, please elaborate.
Thank you for this comment, we improved this fact as highlighted in the text.
In addition, we have closely revised the whole manuscript and sent it to a professional proofreading and editing services to improve language quality. Therefore, we hope, that we have considered all of your suggestions in an adequate way and thank you again for your comments, which helped us to improve the quality of our manuscript substantially.
Thank you again for your comments that greatly helped us to improve our manuscript.
Reviewer 2 Report
In this report the Authors conducted a revision of the literature on adrenocortical carcinoma.
This is a comprhensive work that covers a lot of aspects of this rare tumors in children.
Although this paper deserves publication for its scientific content, the form of the presentation needs substantial improvement. Both sentence construction and spelling must be extensively revised as they have a negative impact on the percieved quality of the presented work.
Author Response
Response to reviewers’ comments
Dear reviewers,
thank you all for your considerations and recommendations that greatly helped us to improve the quality of our manuscript for publication in Cancers.
Please find detailed response to your comments highlighted in the text below.
Review 2
In this report the Authors conducted a revision of the literature on adrenocortical carcinoma.
This is a comprehensive work that covers a lot of aspects of this rare tumors in children.
Although this paper deserves publication for its scientific content, the form of the presentation needs substantial improvement. Both sentence construction and spelling must be extensively revised as they have a negative impact on the percieved quality of the presented work.
Thank you very much for your consideration. We have closely revised the whole manuscript and sent it to a professional proofreading and editing services to improve language quality and think that we could improve the quality of the manuscript substantially.
Thank you again for your comments that greatly helped us to improve our manuscript.
Reviewer 3 Report
Although the title of this article includes “systematic review”, the focus of research objectives is obscure and therefore it is hard for readers to understand the contents of paper. A mere document of previous research would not provide no useful information to readers. Especially, the Tables are too busy and painful for readers, and many grammatical errors are found in manuscript. The authors should prepare more summarized Tables and brush-up their manuscript before submission.
Author Response
Response to reviewers’ comments
Dear reviewers,
thank you all for your considerations and recommendations that greatly helped us to improve the quality of our manuscript for publication in Cancers.
Please find detailed response to your comments highlighted in the text below.
review 3
Although the title of this article includes “systematic review”, the focus of research objectives is obscure and therefore it is hard for readers to understand the contents of paper. A mere document of previous research would not provide no useful information to readers. Especially, the Tables are too busy and painful for readers, and many grammatical errors are found in manuscript. The authors should prepare more summarized Tables and brush-up their manuscript before submission.
Thank you for your recommendation. The goal of the systematic review was to identify knowledge gaps and potential areas for future research as a first step. However, we share your comment that the tables have been too busy and should be more summarized to catch the key messages. Therefore we put the tables containing all details to supplemental material. The key messages are represented in newly structured tables. We have also closely revised the whole manuscript and sent it to a professional proofreading and editing services to improve language quality. Therefore, we think, that we have addressed your recommendations in an adequate way and thank you again for your comments, which helped to improve the quality of our manuscript substantially. Regarding tables and figures following changes have been made:
- Tables 1/2c/3/6 and Figures 1-3 received minor changes and are now entitled Table 1 (remained Table 1), Table 3 (previously Table 2c), Table 4 (previously Table 3), and Table 6 (remained Table 6).
- Tables 2a and b/ 4/ 5/ 7/ 8 have been moved to the supplementary material and are now entitled Suppl. Table 1 a and b (previously Table 2 a and b), Suppl. Table 2 (previously Table 4), Suppl. Table 3 (previously Table 5), Suppl. Table 4 (previously Table 7), and Suppl. Table 5 (previously Table 8).
- Contents of (old) Table 2a and b, Table 4, and Table 8 are now summarized in the new Table 2 and summarized data of Table 7 can be found in the new Table 5.
Thank you again for your comments that greatly helped us to improve our manuscript.
Round 2
Reviewer 1 Report
The authors have significantly improved the manuscript.
However, one major point of concern that remains are the missing methods of the systematic review. The review cannot be independently repeated if the exact search strategy is not included. The prospero ID listed does not work. In addition, referencing according to prospero should be as follows: The protocol for this systematic review was registered on PROSPERO (Unique ID number) and is available in full on the Name of organisation website (URL).
Minor comment: liquid biopsies are not tumor markers, what type of markers would you assess using liquid biopsies?
Author Response
Dear reviewer!
Thank you all for your considerations and recommendations and giving us again the chance to improve our manuscript
Please find detailed response to your comments highlighted in the text below.
Reviewer 1:
However, one major point of concern that remains are the missing methods of the systematic review. The review cannot be independently repeated if the exact search strategy is not included. The prospero ID listed does not work. In addition, referencing according to prospero should be as follows: The protocol for this systematic review was registered on PROSPERO (Unique ID number) and is available in full on the Name of organisation website (URL).
Thank you again for your methodical advices. As recommended we included all necessary data about the exact search strategy in the methods. As highlighted in green art the manuscript:
("child*"[Title/Abstract] OR "pediatric*"[Title/Abstract]) AND ("cancer"[Title/Abstract] OR "carcinoma"[Title/Abstract] OR "tumor"[Title/Abstract] OR "malign*"[Title/Abstract]) AND ("adrenocortical"[Title/Abstract] OR "acc"[Title/Abstract] OR "adrenal*"[Title/Abstract])
Here you find the link of our search for pubmed:
https://pubmed.ncbi.nlm.nih.gov/?term=%28%22child*%22%5BTitle%2FAbstract%5D+OR+%22pediatric*%22%5BTitle%2FAbstract%5D%29+AND+%28%22cancer%22%5BTitle%2FAbstract%5D+OR+%22carcinoma%22%5BTitle%2FAbstract%5D+OR+%22tumor%22%5BTitle%2FAbstract%5D+OR+%22malign*%22%5BTitle%2FAbstract%5D%29+AND+%28%22adrenocortical%22%5BTitle%2FAbstract%5D+OR+%22acc%22%5BTitle%2FAbstract%5D+OR+%22adrenal*%22%5BTitle%2FAbstract%5D%29
The protocol for this systematic review was registered on PROSPERO. Unfortunately PROSPERO. did not proceed with our systematic review registration yet (and we have only a preliminary number as mentioned) and will not do it within the next week due to an overload of COVID-19 related studies, which will be preferentially handled. If a registration is absolutely mandatory for acceptance, we will try to registrate our work within another data base, for example research registration.com, which certainly will take some time.
Minor comment: liquid biopsies are not tumor markers, what type of markers would you assess using liquid biopsies?
We apologize for the lack of accuracy You are totally right that liquid biopsies are not tumor markers in the narrow sense. What we tried to say, liquid biopsies may also be used to validate the efficiency of a cancer treatment and to monitor relapse We changed this passage as highlighted.
As to the English language style and quality we sent the manuscript to a professional proof reading as recommended.
Thank you again for your comments that greatly helped us to improve our manuscript.
Reviewer 3 Report
The authors have appropriately amended their manuscript and the manuscript (especially Tables) was significantly improved. The quality of revised version of their manuscript is enough for publication. I have no further comment.
Author Response
Thank you for your nice comment. We have revised our manuscript further as highlighted due to the comments of reviewer 1 and hope it could further improve the quality of our manuscript.
Round 3
Reviewer 1 Report
Please provide the Embase search strategy as well.
Author Response
Thank you all for your considerations and recommendations and giving us again the chance to improve our manuscript. We have included the embase criteria as well in the method section as suggested. We highlighted the changes in the manuscript.